# Contrastive Vision-Language Learning with Paraphrasing and Negation

## Abstract

Contrastive vision-language models continue to be the dominant approach for image and text retrieval. Contrastive Language-Image Pre-training (CLIP) trains two neural networks in contrastive manner to align their image and text embeddings in a shared latent space. Recent results evaluating CLIP on negated or paraphrased text have shown mixed performance because negation changes meaning radically with minimal lexical changes, while paraphrasing can create very different textual expressions with the same intended meaning. This poses a significant challenge for improving the evaluation results and alignment of vision-language models. To address this challenge, this paper evaluates the combination of paraphrasing and negation, proposes a new CLIP contrastive loss function accounting for both paraphrasing and negation, and applies LLM-generated training triples consisting of original, paraphrased and negated textual captions to CLIP-like training models. The approach, called SemCLIP, is shown to move paraphrased captions towards the original image embeddings while pushing negated captions further away in embedding space. Empirically, SemCLIP is shown to be capable of preserving CLIP's performance while increasing considerably the distances to negated captions. On the CC-Neg benchmark using an *original over negation* image-retrieval accuracy metric, SemCLIP improves accuracy from 68.1% to 78.1%. Although results are mixed when compared with CLIP on the Sugarcrepe++ benchmark, SemCLIP's performance is generally better than the models trained with negated captions. This robustness to negation extends to downstream zero-shot classification tasks where SemCLIP pre-trained on Sugarcrepe++ performs better than CLIP on all tested downstream tasks. These results indicate that SemCLIP can achieve significant robustness to semantic transformations.

## 1 Introduction

We investigate textual semantic similarity in the context of multimodal representation learning. The dominant approach for image and text learning is based on Contrastive Language-Image Pre-training (CLIP) (Radford et al., 2021), a modular architecture where two networks processing images and text, respectively, are trained in contrastive manner to align their embeddings into a shared latent space given image-text pairs. Despite CLIP's considerable practical success, it is well-known that negating the meaning of a textual caption, e.g. by inserting terms such as "not" or "without" often fails to produce the desired change in image-text matching. This is not surprising because such minor lexical changes, though semantically significant, produce embedding vectors that remain close to the original captions in representation embedding space. This reveals a misalignment during CLIP model training between lexical form and semantic content.

Many attempts have aimed to refine learned embeddings to better account for semantics or capture the conceptual relationships in image and text (Fan et al., 2023; Patel et al., 2024; Higgins et al., 2018). Most of these are either application domain-specific or use pre-defined concepts such as *color* and *shape* to disentangle embeddings. In this paper, we adopt a different approach. We map learned embeddings onto a low-dimensional projection within the network structure tasked with learning semantic transformations more generally, that is, learning how meaning is preserved under paraphrasing and inverted under negation.

Multimodal learning is arguably the most promising frontier for the application of learning and reasoning with neural networks, the research goal in the area of neurosymbolic AI (d'Avila Garcez & Lamb, 2020). While language-based reasoning alone is highly ambiguous and context dependent, vision-language models can ground textual descriptions in visual content, allowing one modality to help disambiguate the other. From a neurosymbolic perspective, CLIP-based learning is especially interesting due to its modular architecture, with separate encoders for text and image. However, large multimodal models that combine vision and language have often underperformed relative to pure language models, possibly because logical errors are more easily exposed in a multimodal setting, as seen frequently with the use of negation. This motivates our exploration of enhancing CLIP with complex semantic transformations - contradiction (via negation) and equivalence (via paraphrasing) - and to investigate robustness under such transformations on downstream classification tasks.

As an example, consider the simple task of recognizing *left* and *right*, *above* and *below* in a typical scene understanding scenario using image-caption pairs. By taking the image as being the source of truth, that is, assuming that there are no visual illusions at play, a sentence such as *a blue square left of a red triangle* can be checked for validity. If this sentence is *true* then the sentence *a red triangle right of a blue square* ought to be *true*. Moreover, the sentence *a blue square right of a red triangle* ought to be *false*, assuming no other objects are in the scene. These examples illustrate the difficulty that CLIP-based models face in detecting small lexical but semantically critical changes as well as meaning-preserving changes.

To address the well-known gap between lexical and semantic alignment, CoN-CLIP (Singh et al., 2024) enhanced contrastive learning by incorporating synthetic hard negatives that explicitly encode negation. This approach was motivated by Yuksekgonul et al. 2023 which showed the difficulty of CLIP in handling negation and had applied *data augmentation* approach, referred to as NegCLIP, to automatically generate semantically negated counterparts and trained or finetuned CLIP to distinguish these from the original texts. We adopt this strategy as inspiration for learning a more robust representation of negation.

Kim et al. (2024) highlighted the limitation of CLIP at dealing with linguistic variations such as paraphrasing with their negative impact on tasks that require handling variations of user queries. They proposed ParaCLIP to address this problem by automatically generating two stages of paraphrasing from web-scale image captions using large language models (LLMs), first to convert captions into plain language and then to rephrase the result with similar vocabulary. By finetuning only the text encoder using paraphrases, they achieved substantial improvements on various retrieval tasks in comparison with CLIP. We will investigate negation together with paraphrasing. To the best of our knowledge, this is the first paper to study both challenges together.

The above findings point to the benefit of text-image data augmentation for contrastive learning, while also highlighting the need to account for different lexical forms and semantic changes. We seek to achieve this by introducing a new embedding projection in the CLIP architecture through a new loss function with pairwise components for negation and paraphrasing with respect to the original text, in addition to CLIP's original contrastive loss for image-caption pairs. The goal is to maintain performance on the original data while improving robustness to semantic variations. We refer to this architecture and associated training loss as SemCLIP which extends CLIP by enabling joint learning of negation and paraphrasing. The two new loss function components added to the original contrastive loss in CLIP are: (1) Paraphrasing Loss, $L_{paraphrase}$, encouraging semantic invariance in the case of captions that differ syntactically but have very similar meaning; paraphrased captions will be mapped near to the original caption and image in embedding space; (2) Negation Loss, $L_{negation}$, enforcing semantic exclusivity on text that explicitly contradicts the original caption; negated captions will be mapped far from the original caption and corresponding image in embedding space.

By jointly optimizing $L_{paraphrase}$ and $L_{negation}$ alongside the standard CLIP contrastive loss, $L_{contrastive}$, our results show that SemCLIP can preserve the image retrieval performance of CLIP on original and paraphrased captions while significantly increasing the accuracy of original captions over negated captions, as defined in Singh et al. 2024. Our experimental evaluation indicates that, when combining negation and paraphrasing, SemCLIP achieves robustness to paraphrased queries (defined precisely in what follows), while maintaining a reasonable performance on standard retrieval tasks.

The remainder of the paper is organized as follows: related work is discussed in the next section; the SemCLIP architecture, proposed loss function and evaluation methodology are then introduced; experimental findings are presented and discussed, followed by the conclusion and future work.

## 2 RELATED WORK

This section reviews prior work addressing the challenges of negated and paraphrased captions on vision language model (VLM) training, and situates our contribution in the context of relevant work and evaluation benchmarks.

The effectiveness of VLMs has been limited by a lack of semantic understanding. Yuksekgonul et al. (2023) showed that many VLMs behave as *bags-of-words* overly sensitive to specific keywords. They introduced the Attributes, Relations and Orders (ARO) benchmark test cases to quantitatively evaluate attribute binding, relational understanding and order sensitivity. The evaluation revealed near-chance performance even from state-of-the-art VLMs, and they proposed *composition-aware hard negative mining* as a solution. Hsieh et al. (2023) later identified that generated negative captions in ARO were often flawed (grammatically incorrect or lacking a plausible visual context).

To counter this vulnerability, Hsieh et al. (2023) introduced *Sugarcrepe*, intended to be a more robust benchmark dataset, that leveraged Large Language Models (LLMs) to generate more fluent and plausible negative captions, also applying adversarial refinements to mitigate annotation bias. Their findings revealed that the performance of many VLMs had been overstated, underscoring the complexity of accurately evaluating compositionality and semantics of VLMs. Building on this, Sugarcrepe++ (Dumpala et al., 2024) extended the benchmark by incorporating a deeper analysis of model sensitivity to both lexical and semantic variations. While Sugarcrepe focused on distinguishing between correct captions and lexically similar but semantically different negatives, it did not account for semantically equivalent but lexically different paraphrases. Sugarcrepe++ addressed this by associating each image with a triplet of captions: two semantically equivalent but lexically different positive captions, and one hard negative caption. Evaluations using Sugarcrepe++ revealed that strong performance on traditional compositionality benchmarks does not translate necessarily to success on this more fine-grained benchmark. Many models struggled with semantic coherence when object attributes or spatial relations were changed. Improving semantic invariance, i.e. robustness to lexical variation without semantic change remained a challenge, and it is one of the goals of this paper. This task is harder than investigating negation alone, as detailed in our experimental evaluation. We therefore exercise caution when making direct comparisons with existing approaches that do not account for both negation and paraphrasing.

Recently, LaCLIP (Fan et al., 2023) addressed the data augmentation asymmetry in CLIP by training using a random mix of original captions and diverse variations of LLM-generated *rewrites*. Para-CLIP (Kim et al., 2024) introduced a more targeted paraphrasing generation using an LLM to clean up noisy captions, generating better paraphrasing data to finetune the text encoder. They showed that it is possible to enhance the performance of CLIP models by including retrieval with paraphrasing. LLip (Lavoie et al., 2024) challenged CLIP's assumption of a single image embedding, proposing instead a set of visual mixture tokens combined via multi-head attention to match diverse captions. DreamLIP (Zheng et al., 2024) explored the semantic richness of long, detailed captions. With the understanding that each sentence in the long caption often described a partial aspect of the image, DreamLIP sampled sub-captions to construct multiple positive pairs against image patches. It introduced a grouped subcaption-specific loss to match the embedding of these sub-captions with corresponding local image patches. This fine grained alignment enabled better performance over CLIP without requiring significantly more data. LLip and DreamLIP showed the viability of associating an image with diverse textual descriptions. Our work takes inspiration from that, although by contrast with the above approaches our method seeks to preserve CLIP's modularity.

Another line of research has been focused on enforcing semantic exclusivity for negation, that is, training CLIP models to push apart embeddings that are semantically opposite even when they share high lexical similarity. CoN-CLIP (Singh et al., 2024) introduced the CC-Neg dataset which paired images with their corresponding original captions as well as negated captions generated by an LLM. Negation terms are inserted into the subject and predicate-object pairs. Their modified contrastive loss explicitly separated image embeddings from the embeddings of those negated captions. This inspired our formulation of a negation loss. Similarly, TripletCLIP (Patel et al., 2024) generated

fully synthetic vision-language negative pairs: an LLM created hard negative captions and a text-to-image model synthesized corresponding negative images. Training then used a triplet loss to align positive image-caption pairs and push away negative pairs.

Each of the above work has shown improvement over CLIP's performance using either paraphrased captions or negated captions in isolation, sometimes showing substantial performance gains depending on the choice of datasets. Our work builds on such papers but also addresses a harder problem: comprehensive performance improvements across the semantically diverse space of paraphrasing and negation together, maintaining CLIP's modularity and adopting a simple embedding projection, as detailed in the next section.

## 3 METHODOLOGY: SEMANTIC CLIP WITH PARAPHRASING AND NEGATION

Our method extends CLIP (Radford et al., 2021) by engineering a new loss function to incorporate the concepts of paraphrasing and negation. The objective is to enrich and evaluate the joint embedding space with such concepts. We expect that this new representation of negation, contrasting negation with an original caption but also with other ways of stating that original caption (paraphrasing), will produce a more robust semantic alignment between text and image. By *robust* we mean a representation capable of distinguishing small changes in text that may invert its meaning, and considerable changes in text that will nevertheless maintain the original meaning. This will be evaluated on downstream image classification tasks in comparison with the closest related work investigating CLIP with negation (Singh et al., 2024) and with paraphrasing (Kim et al., 2024). As mentioned, to the best of our knowledge, no other work has investigated the combination of the two.

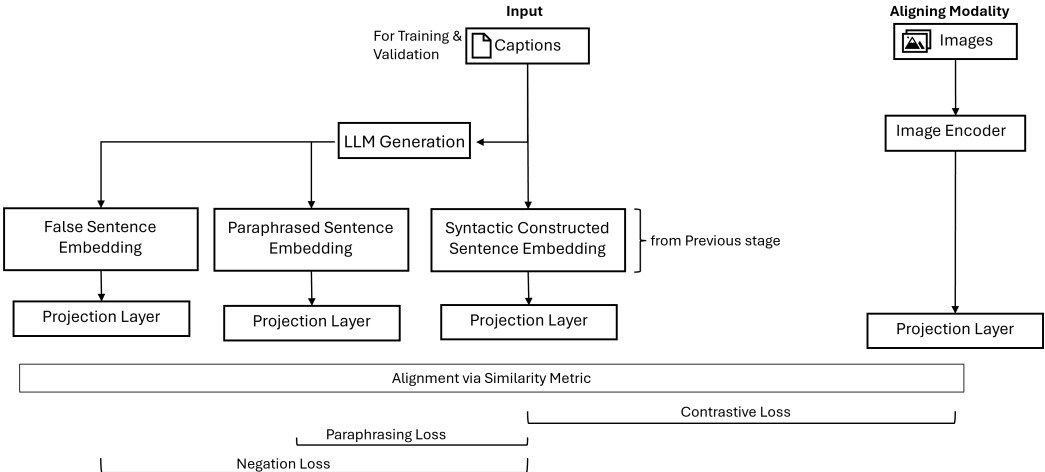

Figure 1: SemCLIP architecture including the contrastive loss $L_{contrastive}$, paraphrasing loss $L_{paraphrase}$ and negation loss $L_{negation}$ showing how the embeddings of image-caption pairs are aligned through the proposed embedding projection and new training loss function (Eq. 7).

### 3.1 CAPTION AUGMENTATION

Given an image-text pair $(I, c)$, where $c$ is the textual caption description of image $I$, SemCLIP generates two additional captions:

- a paraphrased caption $c^+$, preserving the meaning of $c$ but differing in lexical or syntactic structure;
- a negated caption $c^-$, representing the opposite or logical negation of the original description.

Given a CLIP model, text embeddings can be obtained from $c$, $c^+$ and $c^-$, denoted respectively as $\mathbf{t}$, $\mathbf{t}^+$, and $\mathbf{t}^-$. All embeddings are $\ell_2$-normalized such that $\|\mathbf{t}\|_2 = \|\mathbf{t}^+\|_2 = \|\mathbf{t}^-\|_2 = 1$.

## 3.2 Embedding Projections

Instead of disentangling trained embeddings to identify concepts and their negation, which is very challenging (Higgins et al., 2018), we define a set of projection directions in the embedding space and learn to order embeddings based on those projections, aiming to separate $\mathbf{t}^+$ from $\mathbf{t}^-$. We initialize $n$ orthonormal vectors $\mathbf{v}_i \in \mathbb{R}^d$, $\quad i = 1, \ldots, n, \|\mathbf{v}_i\|_2 = 1$, where $d$ is the dimensionality of the embedding space. These vectors are sampled from a standard normal distribution, made orthogonal via the Gram-Schmidt process (Leon et al., 2013), and $\ell_2$-normalized. Given a unit-length embedding vector $\mathbf{t} \in \mathbb{R}^d$, we compute its scalar projection onto each $\mathbf{v}_i$ by the dot product:

$$p_i(\mathbf{t}) = \mathbf{t}^\top \mathbf{v}_i. \tag{1}$$

The full projection of $\mathbf{t}$ onto the subspace spanned by $\{\mathbf{v}_i\}_{i=1}^n$ is represented by the vector of scalar projections:

$$\mathbf{p}(\mathbf{t}) = \begin{bmatrix} p_1(\mathbf{t}) \\ \vdots \\ p_n(\mathbf{t}) \end{bmatrix} \in \mathbb{R}^n. \tag{2}$$

We can alternatively express the projection by stacking the orthonormal vectors into a matrix $V = \begin{bmatrix} \mathbf{v}_1 & \mathbf{v}_2 & \cdots & \mathbf{v}_n \end{bmatrix} \in \mathbb{R}^{d \times n}$. Then, the vector of scalar projections is given by the matrix multiplication:

$$\mathbf{p}(\mathbf{t}) = V^\top \mathbf{t} \in \mathbb{R}^n, \tag{3}$$

where each component corresponds to $p_i(\mathbf{t}) = \mathbf{v}_i^\top \mathbf{t}$.

Optionally, $\mathbf{p}(\mathbf{t})$ can be $\ell_2$-normalized to maintain consistent scale between projections, and projection vectors $\{\mathbf{v}_i\}$ may be updated during training by gradient descent.

The projection subspace dimension $n$ is chosen to be much smaller than the original embedding dimension to encourage a more interpretable, low-dimensional semantic representation. Rather than simply performing dimensionality reduction, the goal is to define a subspace where semantic relations for paraphrasing and negation correspond to meaningful geometric constraints on the projected embeddings. The projections onto the subspace are intended to provide a structured interpretation of the semantic relationships between two captions, with negation and paraphrasing modeled as constraints in the subspace, as follows.

## 3.3 Paraphrasing and Negation Losses

Our training objective extends CLIP's original contrastive loss by introducing two new components: a *paraphrasing loss* to enforce consistency across paraphrased captions, and a *negation loss* to repel negated captions in projection space. Let $\{\mathbf{i}_i\}_{i=1}^N$ and $\{\mathbf{t}_i\}_{i=1}^N$ denote batches of $\ell_2$-normalized image and text embeddings, respectively. The similarity scores between all image-text pairs in the batch are computed as: $S_{ij} = \tau \cdot \cos(\mathbf{i}_i, \mathbf{t}_j)$, where $\tau = \exp(\theta)$ is a learnable temperature parameter.

The standard CLIP contrastive loss is then given by:

$$\mathcal{L}_{\text{contrastive}} = \frac{1}{2N} \sum_{i=1}^N \left( \mathcal{L}_{\text{CE}}(S_{i,:}, i) + \mathcal{L}_{\text{CE}}(S_{:,i}, i) \right), \tag{4}$$

where $\mathcal{L}_{\text{CE}}(\mathbf{v}, i)$ denotes the cross-entropy loss with logits $\mathbf{v}$ and ground-truth class $i$, while $S_{i,:}$ and $S_{:,i}$ represent the $i$-th row and column of the similarity matrix $S$, corresponding to image-to-text and text-to-image similarities, respectively.

We define two additional loss-function components: the *paraphrasing loss* encourages the projections of an original caption and its paraphrase to lie in the same region of the subspace, while the *negation loss* encourages the projections of a caption and its negation to point in different directions:

$$\mathcal{L}_{\text{paraphrase}} = 1 - \cos(\mathbf{p}(\mathbf{t}), \mathbf{p}(\mathbf{t}^+)) \tag{5}$$

$$\mathcal{L}_{\text{negation}} = \max\left(0, \cos(\mathbf{p}(\mathbf{t}), \mathbf{p}(\mathbf{t}^-))\right) \tag{6}$$

with cosine similarity given as usual by $\cos(\mathbf{a}, \mathbf{b}) = \frac{\mathbf{a}^\top \mathbf{b}}{\|\mathbf{a}\|_2 \|\mathbf{b}\|_2}$.

The losses encourage paraphrased embeddings $\mathbf{t}^+$ to project in the same direction as the original $\mathbf{t}$ (cosine similarity close to 1), while negated embeddings $\mathbf{t}^-$ are encouraged to project in an orthogonal or dissimilar direction (cosine similarity less than or equal to zero). Thus, the relation between original, paraphrased, and negated captions is captured via angular separation in this subspace.

The combined loss function (Eq. 7) aims to capture the essential semantic dichotomy between paraphrasing (semantic equivalence) and negation (semantic opposition) through a simple geometric constraint. Implicitly, this is encoded at least in one direction in the projection subspace. When $n = 1$ we have simply a binary signal (or bit) distinguishing paraphrasing (on) from negation (off), or vice-versa. In other words, the subspace projections attempt to summarize the semantic relations as a discrete factor to facilitate interpretation and downstream tasks. The overall loss function combines all three components:

$$\mathcal{L}_{\text{total}} = \frac{\alpha \mathcal{L}_{\text{contrastive}} + \beta \mathcal{L}_{\text{paraphrase}} + \gamma \mathcal{L}_{\text{negation}}}{\alpha + \beta + \gamma} \tag{7}$$

**Hyperparameters:** In the experiments to follow, for simplicity and ease of analysis, we use $\alpha, \beta, \gamma \in \{0, 1\}$ only. Future work with a focus on the optimization of our results may consider a fine-grained grid search over $\alpha, \beta, \gamma$, or the use of learned weights for the components of the loss. Some additional hyperparameters control the behavior of the projection mechanism: (a) *num_projection_vectors (n)*: number of projection directions used; (b) *normalize_projections* (Bool): whether to $\ell2$-normalize the projections $\mathbf{p}(\mathbf{t})$ prior to the loss computation.

For simplicity, all our experiments use $n \in \{1, 2\}$. This should provide intuitive visualizations to help interpret or debug the learned projections, although we do not investigate this in detail in this paper. Exploring other values for $n$ is of course possible and left as an interesting direction for future work, investigating how projection dimensionality affects semantic representation or disentanglement. We also tested on projection normalization in all the experiments reported here. The projection vectors can be learnable too, although our experiments did not show an improvement with learnable projections. See Appendix C for ablation study on these hyperparameters.

**Architecture:** Figure 1 illustrates the SemCLIP model and application of the combined loss function (Eq. 7). We follow the CLIP architecture outlined in Radford et al. (2021), which is composed of two key components: a vision encoder and a transformer-based text encoder. We augment it only by incorporating the shared projection space defined above. We shall now discuss in more detail about our CLIP implementation.

*Vision Encoder:* We adopt a Vision Transformer (ViT-B/32) as the vision encoder, initialized with pretrained weights from LAION-2B (Cherti et al., 2023). Our code adapts the standard OpenCLIP implementation from Ilharco et al. 2021. We use the pretrained OpenCLIP models with frozen vision encoders during training.

*Text Encoder:* The text encoder is a Transformer with learnable token embeddings and positional encoding. By default, it consists of $L = 12$ layers with $H = 8$ attention heads and uses a hidden dimension of $d = 512$. The encoder operates over tokenized input captions, as usual, and the final output is derived from the representation at the end-of-sequence (EOS) token position. This representation is then passed through a linear projection layer to align with the dimensionality of the vision encoder output.

## 4 EXPERIMENTAL EVALUATION

We evaluated SemCLIP using CC-Neg (Singh et al., 2024) and Sugarcrepe++ (SCPP) (Dumpala et al., 2024) motivated by the Related Work. Our goal is to align the image-caption pairs and evaluate performance as a proxy for semantic understanding. Thus, we also selected five datasets for a downstream zero-shot image classification task, namely CIFAR10 and CIFAR100 (Krizhevsky, 2009), Flowers 102 (Nilsback & Zisserman, 2008), Food 101 (Bossard et al., 2014) and Oxford-IIIT Pet (Parkhi et al., 2012).

**Datasets:** CC-Neg has been used to evaluate negation understanding in vision-language models. Based on the CC-3M corpus, it contains 228,246 image-caption pairs accompanied by negated cap-

Table 1: Evaluation of CLIP (baseline), CLIP with paraphrasing, CLIP with negation and Sem-CLIP (ours) on the CC-Neg and Sugarcrepe++ (SCPP) datasets. The table shows Top-1 accuracy on image-caption matching for the original and paraphrased captions, ratio between the accuracies on original and negated captions, and composite score obtained from Eq.8. On CC-Neg, Sem-CLIP maintains Top-1 accuracy with respect to the CLIP baseline while improving considerably the original-over-negation accuracy, achieving the best composite score on CC-Neg. Although results are mixed in the case of the much smaller Sugarcrepe++ dataset, contrast the performance degradation of CLIP with Negation with respect to CLIP Baseline (up to 10 percentage points) with that of SemCLIP w.r.t. CLIP Baseline. SemCLIP is clearly less sensitive to negation.

| Metric | Dataset | CLIP Baseline | Paraphrase only | Negation only | SemCLIP (ours) |
|---|---|---|---|---|---|
| Original Caption (Top-1 Acc) | CC-Neg | 33.1 | 33.0 | 33.0 | 33.1 |
| | SCPP | 64.2 | 63.6 | 53.3 | 57.3 |
| Paraphrased Caption (Top-1 Acc) | CC-Neg | 21.9 | 23.0 | 20.1 | 21.0 |
| | SCPP | 60.0 | 59.1 | 46.7 | 53.1 |
| Original over Negated (Acc) | CC-Neg | 68.1 | 66.8 | 75.6 | 78.1 |
| | SCPP | 82.7 | 82.3 | 80.8 | 82.8 |
| Composite Score | CC-Neg | 30.4 | 29.9 | 34.8 | 36.8 |
| | SCPP | 63.2 | 62.4 | 53.9 | 58.6 |

tions using terms such as "no", "not" and "without". CC-Neg was used by others to show how language models often mis-align negated captions with images. Sugarcrepe++ is a curated dataset used to evaluate compositional understanding of text-to-images mappings with a total of 4,757 image-caption pairs whereby object attributes or their relations are swapped but semantic similarity is maintained. CIFAR-10 is the standard computer vision benchmark containing 60,000 low-resolution 32x32 color images equally distributed across 10 mutually-exclusive object classes. CIFAR-100 has the same 60,000 images but classified across 100 more fine-grained sub-classes of the original 10. Food 101 contains 101,000 images classified into 101 food categories. With images collected from the web, the data is not cleaned-up and some images also have wrong labels. Food 101 is widely used to benchmark model robustness. Flowers 102 contains 8,189 images of 102 commonly found flower species. It has significant variation in lighting and scale and visually similar flowers from different classes. Oxford-IIIT Pet contains 37 breeds of cats and dogs with approximately 200 images per class and presenting similar challenges as Flowers 102 on scale and pose.

**Synthetic Caption Generation:** To address the need for large quantities of paraphrased and negated captions, we employ a two stage LLM-based synthetic caption generation. First, a Phi-4 model (Abdin et al., 2024) generates candidate paraphrased and negated captions from original captions. A second model, Mistral-7B (Jiang et al., 2023), provides an independent validation by assessing the quality of the generated captions. When a candidate fails, the validation provides an alternative. Using two models with distinct training and architectures is expected to reduce correlated errors. Prompt templates for both generation and validation are presented in Appendix A. The pipeline is implemented with the Ollama package (Ollama, 2023) for straightforward model invocation, and the inference temperature is set to 0.1 to reduce stochastic variation during caption generation.

**Training Configuration:** As detailed in the methodology section, image-caption pairs (together with their paraphrased and negated synthetic counterparts) are trained using contrastive learning. Training is carried out with the AdamW optimizer (Loshchilov & Hutter, 2017a) ($\beta_1 = 0.9$, $\beta_2 = 0.999$, Weight decay: 0.2) for 200 epochs and at a learning rate of $5 \times 10^{-5}$. The learning rate follows a linear warmup over 50 steps and then adopts a cosine annealing schedule (Loshchilov & Hutter, 2017b) without warm restarts. The number of projection layers, normalization of projections and the use of learnable projections (see Appendix B for details), are also evaluated in an ablation test, with those results reported in Appendix C. Gradients are accumulated over two steps, and gradient clipping is applied with a maximum norm threshold of 1.0.

**Experimental Findings:** Table 1 summarizes the results based on 25 individual training runs that systematically sweep the hyperparameters listed under Training Configuration. Four variants of

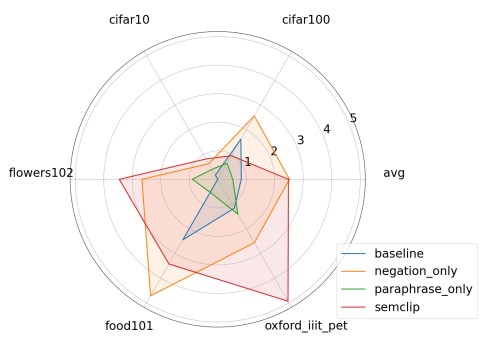 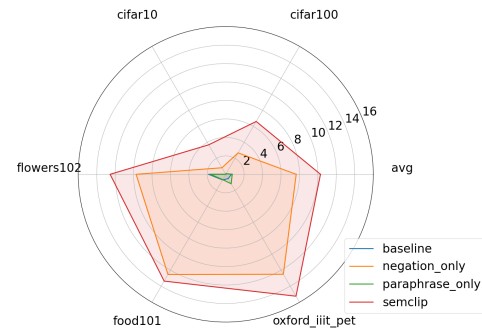

(a) Mean Accuracy Delta (CC-Neg finetuning).      (b) Mean Accuracy Delta (SCPP finetuning).

Figure 2: Model robustness to negation is measured by the difference in accuracy (Mean Accuracy Delta) between positive and negative captions, e.g. "This is not a photo of a <class>." Any negative Delta is shown as zero. SemCLIP achieves the highest Delta on three of five downstream datasets when finetuned on CC-Neg (a) and on all datasets when finetuned on Sugarcrepe++ (SCPP) (b); further detailed evaluations are reported in Appendix C.

CLIP were compared on the CC-Neg and Sugarcrepe++ (SCPP) datasets, each using different loss function terms, including the proposed SemCLIP model, as follows:

- **CLIP Baseline** – baseline model with only contrastive loss ($\beta = 0$ and $\gamma = 0$ in $\mathcal{L}_{\text{total}}$).
- **Paraphrase_only** – adds the paraphrase loss $L_{\text{paraphrase}}$ to the contrastive loss ($\gamma = 0$ in $\mathcal{L}_{\text{total}}$).
- **Negation_only** – adds the negation loss $L_{\text{negation}}$ to the contrastive loss ($\beta = 0$ in $\mathcal{L}_{\text{total}}$).
- **SemCLIP** – combines both $L_{\text{paraphrase}}$ and $L_{\text{negation}}$ with the contrastive loss.

Table 1 shows the standard Top-1 accuracies, the original-over-negation accuracy, which measures whether an original caption is closer in embedding space to its corresponding image than its corresponding negated caption, as proposed in Singh et al. (2024), and a composite score that evaluates overall performance across original, paraphrased and negated image-caption pairs, defined as:

$$\text{Composite Score} = \frac{1}{3} \left( Acc_{\text{orig}} + Acc_{\text{para}} + \tilde{Acc}_{\text{neg}} \right) \tag{8}$$

where $Acc_{orig}$ is the Top-1 accuracy at image retrieval using the original caption, $Acc_{para}$ is the Top-1 accuracy at image retrieval using the paraphrased caption and $\tilde{Acc}_{neg}$ is the original-over-negation accuracy. Since for this metric a 50% accuracy corresponds to random choice, we re-scale the accuracy $Acc_{neg}$ to range from 0% to 100% using: $\tilde{Acc}_{neg} = \max\left(0, 2(Acc_{neg} - 0.5)\right)$.

All four models recover the correct image for approximately 33% of the 22,825 held-out examples in the CC-Neg test set, that is, showing Top-1 original caption accuracy around 33%. It should be noted that the 33% value may appear low as it refers to an exact match out of all other 22,824 captions. This indicates that the addition of the paraphrasing and negation terms to the loss function in SemCLIP does not degrade the accuracy obtained with the other contrastive models.

The most striking effect of SemCLIP is found with negated captions by including the negation term $L_{negation}$, as measured by the accuracy of the original caption over negation. This metric has shown that models with an explicit loss function component dealing with negation (Negation_only and even more so SemCLIP) can reduce considerably the tendency of CLIP to align an image with its negated caption. In SemCLIP, image embeddings are explicitly aligned more closely to their original caption embeddings than to a negated textual counterpart.

Based on the composite score introduced in this paper to combine the evaluation of negated captions with that of paraphrasing, SemCLIP shows the best performance overall (although only marginally better in the case of SCPP), being capable of aligning simultaneously original and paraphrased captions with corresponding images while reducing the cases where an image is wrongly matched to its negated caption. Following Singh et al. 2024, what is done here in the absence of time consuming user validations, is check that the image is not matched to the negated caption, as measured by the

original-over-negation accuracies. Better accuracy there is expected to produce a better performance on downstream tasks, as will be evaluated using CIFAR, Flowers, Food 101 and Oxford-IIIT Pet.

We observe that direct performance comparisons are not entirely possible since the task proposed in this paper of handling both negation and paraphrasing together is different from addressing them separately. Nevertheless, the composite score introduced here is intended to evaluate the overall balance between learning textual equivalence (paraphrasing) and contradictions (negation). In the case of CC-Neg, there is an improvement in the Composite Score of Negation_only w.r.t. the CLIP baseline and a clear improvement overall with SemCLIP. In the case of the well-curated but much smaller SCPP, the Composite Score of Negation_only is much lower than the CLIP baseline, which is reflected in the overall lower Composite Score of SemCLIP.

All of the above experiments were carried out from scratch running the OpenCLIP implementation extended to include the paraphrasing loss, negation loss and SemCLIP. Comparisons with the results reported in the literature should consider possible variations in implementation. We nevertheless report these here: based on the *original-over-negation* results reported in (Singh et al., 2024) on the CC-Neg finetuning, SemCLIP achieves 78.1% accuracy compared with CLIP's 65.70%, Neg-CLIP's 62.63%, FLAVA's 58.93% (Singh et al., 2021) and BLIP's 62.31% (Li et al., 2022). SemCLIP only achieves a lower accuracy than CoN-CLIP's impressive 99.70% (Singh et al., 2024). To the best of our knowledge, CoN-CLIP has not been evaluated on Sugarcrepe++.

Next, we evaluate downstream zero-shot classification tasks using the CIFAR, Flowers-102, Food-101, and Oxford-IIIT Pet datasets. For each image classification dataset, we generate a caption for every class label using the prompt: "This is a photo of a <class>". Without any further fine-tuning of the CLIP models, each image is encoded by the model and compared against the embeddings of all class captions. The caption whose normalized embedding is closest with respect to cosine similarity to the image's normalized embedding is selected as the predicted class. This yields the Top-1 accuracy for the downstream zero-shot classification task. In addition, we generate negated captions of the form "This is not a photo of a <class>". The same process is repeated using these negated captions to compute a classification accuracy metric under negation. A lower accuracy here indicates a higher misclassification rate, which is the desired behavior when using negated captions. A low accuracy under negation indicates that the model chooses any class other than the original, e.g. if the negated caption is "this is not a photo of a dog" and the image is that of a dog then we are satisfied if the model chooses any other class (e.g."cat").

Following Singh et al. 2024, we compute the difference (Delta) between the accuracy with positive captions and the accuracy under negation to quantify the model's sensitivity to negation. A larger Delta is taken to indicate a greater ability of the model to distinguish between positive and negated language in relation to the image content. Results are shown in Figure 2. Finetuned using CC-Neg, SemCLIP produces, zero-shot, a larger Delta than the other models on 3 out of 5 downstream tasks; finetuned on SPCC, SemCLIP produces zero-shot a larger Delta on all downstream tasks.

## 5 CONCLUSION AND FUTURE WORK

We presented SemCLIP, a method that enhances CLIP's robustness to linguistic variations through joint learning of paraphrasing and negation. By incorporating projection-based paraphrasing and negation losses alongside the standard contrastive objective, our approach addresses key limitations in handling semantic variations while preserving retrieval performance. Based on the empirical results obtained here, we argue that contrastive learning should address paraphrasing and negation together, rather than in isolation.

Much recent research has focused on addressing CLIP's *negation problem*. Broader concerns however include equivalence, contradiction and entailment in natural language inference (Srivastava et al., 2023). Addressing these within contrastive objectives presents major challenges. While not addressing entailment in this paper, analyzing negation and paraphrasing together provides a more realistic setting for learning sound semantics in CLIP-like systems. The paper's main proposition for structuring embedding spaces to reflect paraphrasing (equivalence) and negation (contradiction) should yield representations that are better suited for entailment inference. Future work will extend this approach to more complex relationships, such as non-verbal negation (Vanek & Zhang, 2023) and explore data augmentation strategies to support the implementation of entailment.

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

## A  PROMPT TEMPLATES FOR CAPTION GENERATION

The following templates provide an example of paraphrased and negated caption candidate generation by either Phi-4 or Mistral-7B model as well as the subsequent quality validation of generation by Mistral-7B.

---

**Caption Generation Template**

```
1  Paraphrased caption candidate
2  (Phi-4/Mistral-7B)
3  User: Rephrase the following sentence with the same meaning: {
       original}
4
5  Negated caption candidate
6  (Phi-4/Mistral-7B)
7  User: Generate a completely false version of this sentence: {
       original}
```

---

**Caption Validation Template**

```
1   Paraphrased caption candidate
2   (Mistral-7B)
3   User: The original sentence is: "{original}"
4     The generated paraphrase is: "{paraphrased}"
5     Does the paraphrase accurately retain the meaning of the
          original? Respond with "Yes" or "No".
6
7   Negated caption candidate
8   (Mistral-7B)
9   User: The original sentence is: "{original}"
10    The generated false sentence is: "{false_sentence}"
11    Does the false sentence contradict or misrepresent the original
          ? Respond with "Yes" or "No".
```

## B   IMPLEMENTATION DETAILS

Details of hardware used in this work is as follows:

1. **Platform**: Azure (Standard NC40ads H100 v5)
2. **vCPU**: 40
3. **RAM**: 320GB
4. **GPU**: 1 x Nvidia H100
5. **Operating System**: Ubuntu 22.04

Details of the training hyperparameters used are as follows:

### B.1   IMAGE NORMALIZATION

*OpenAI* normalization values (as implemented via PyTorch transforms in the OpenCLIP library) were used for all training and testing runs.

### FIXED PARAMETERS

1. **Model**: ViT-B-32 (Pretrained: laion2b_s34b_b79k)
2. **Epoch**: 200
3. **Learning Rate**: 5e-5
4. **Learning Rate schedule**: Linear warmup over 50 steps; cosine annealing without warm restarts
5. **Optimizer**: AdamW ($\beta_1$: 0.9, $\beta_2$: 0.999, Weight Decay: 0.2)
6. **Random Seed**: 42

### VARYING PARAMETERS

7. **Number of Projection Vectors**: [1, 2]
8. **Use Learnable Projections**: [True, False]
9. **Projection Normalization**: [True, False]

## C   ABLATION STUDIES AND FURTHER RESULTS

Our first ablation study investigates the effect of key hyperparameters on image matching accuracies specifically: (1) the number of projection vectors, (2) the use of learnable projections, and (3) the normalization of projection vectors. These factors are evaluated across three tasks: matching with original captions, paraphrased captions, and the original over negated caption task.

Figures 3a, 4a, and 5a present results using original captions. All models trained with CC-Neg dataset (Singh et al., 2024), including our proposed SemCLIP, achieve comparable accuracies to the CLIP baseline, consistently around 33.0%. The selected hyperparameters show negligible impact indicating that our modifications preserve the model's core image matching capability using original captions.

By contrast, performance applied to paraphrased captions (Figures 3b, 4b, and 5b) drops, ranging between 20.1% and 23.0%. This decrease highlights the challenge posed by linguistic variation. Among the hyperparameters, the number of projection vectors has the most noticeable influence on SemCLIP with a 1D projection producing higher accuracy than a 2D projection.

The most pronounced hyperparameter effects appear in the original over negated task. As presented in Figures 3c, 4c, and 5c, the CLIP baseline and Paraphrase_only models struggle with accuracies below 50%, suggesting an inability to reliably distinguish between the intended caption and its negation. By contrast, models explicitly trained on negation data (Negation_only and SemCLIP)

perform substantially better. SemCLIP achieves peak accuracy of 89.1%. Other hyperparameters have limited impact on this task across all models.

Repeating the experiment with models pretrained on the Sugarcrepe++ (Dumpala et al., 2024) dataset produce the results shown in Figures 6, 7, and 8. SemCLIP shows consistent performance across all image matching tasks. On original captions, SemCLIP achieves up to 64.6% accuracy (1D projection) and 48.2% (2D projection), while maintaining robustness on paraphrased captions (with performance near to that of using original captions). On the negation task, all models perform well (80.8 - 82.83%), with SemCLIP slightly outperforming the rest.

Overall, these findings highlight SemCLIP's ability to learn robust representations to paraphrasing and negation without much performance degradation on the original task. Among the hyperparameters studied, the number of projection vectors is worth further investigation, offering a potential direction for future fine-tuning and performance optimization.

Analysis of standard accuracy and negated accuracy diagrams on downstream classification tasks: SemCLIP using either dataset has outperformed in accuracy to those reported in CoN-CLIP for CIFAR10 and CIFAR100 but not for Flower102, Food101 and Oxford-IIIT Pet (see Table 2). It may be unrealistic to assume that the combination of negation and paraphrasing should be sufficient to address the difficult task of distinguishing the very similar-looking objects in some of these datasets. Nonetheless, these results have reinforced the significance of the proposed training loss function with paraphrased and negation loss components as a richer way of investigating semantic robustness of image-caption pairs. Figures 9 and 10 presents the classification accuracy on (a) original caption and (b) negated caption, as well as (c) the accuracy delta between the two. SemCLIP has shown an improved robustness to negation overall as it has the largest delta on three of five tasks when trained with CC-Neg dataset and on all tasks when trained with SCPP.

Table 2: Evaluation of zero-shot image classification with reference to Singh et al. 2024. Five downstream tasks are evaluated and the best scores are highlighted in **bold**. Results from CoN-CLIP are extracted from Table 4 in Singh et al. 2024.

| Downstream Tasks | Models | | |
|---|---|---|---|
| | SemCLIP (CC-Neg) | SemCLIP (SCPP) | CoN-CLIP |
| CIFAR-10 | 90.6 | **91.6** | 90.45 |
| CIFAR-100 | 62.5 | **66.5** | 62.31 |
| Foods101 | 50.7 | 62.7 | **83.39** |
| Flowers102 | 36.0 | 37.5 | **64.74** |
| Oxford-IIIT Pet | 53.5 | 69.5 | **81.66** |

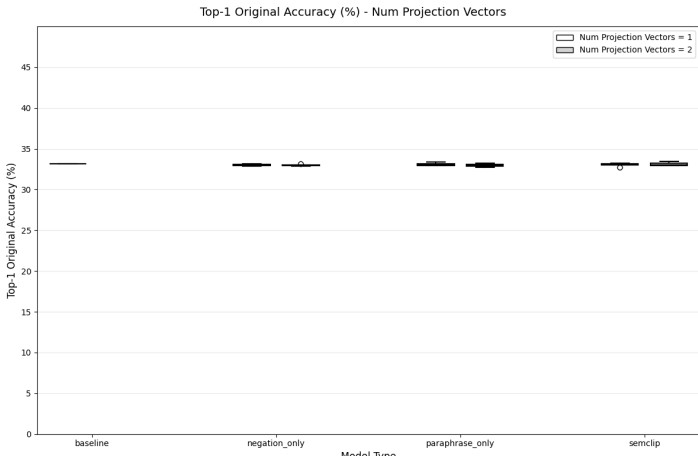

(a) Effect of setting the number of projection vectors on Top-1 accuracy using original caption for image matching.

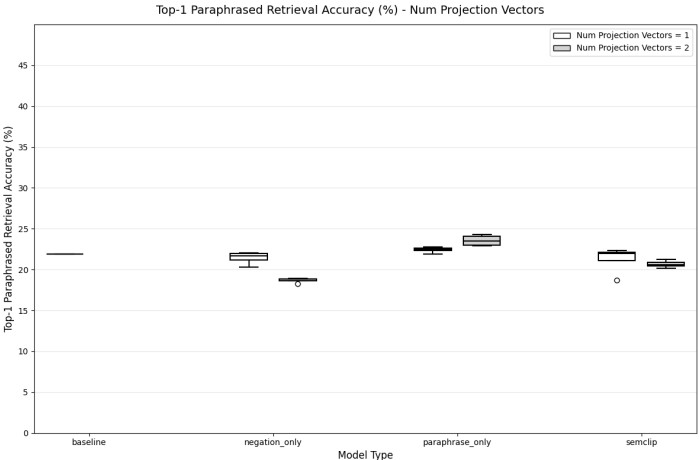

(b) Effect of setting the number of projection vectors on Top-1 accuracy using paraphrased caption for image matching.

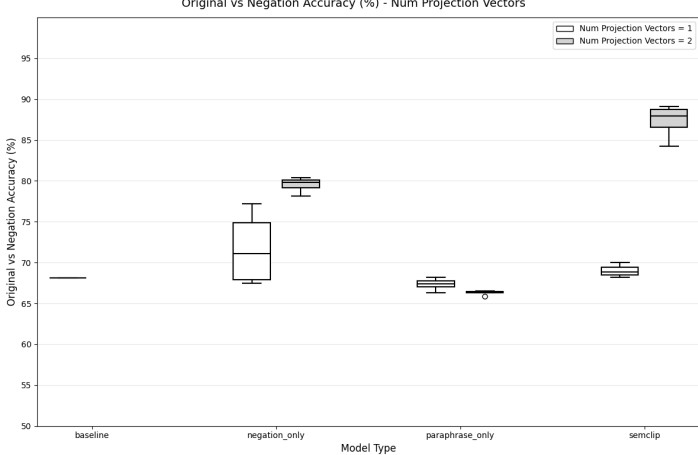

(c) Effect of setting the number of projection vectors on Top-1 accuracy using original caption over negated caption for image matching.

Figure 3: Effect of setting the number of projection vectors on image matching accuracies using trained model (finetuned with CC-Neg dataset (Singh et al., 2024)).

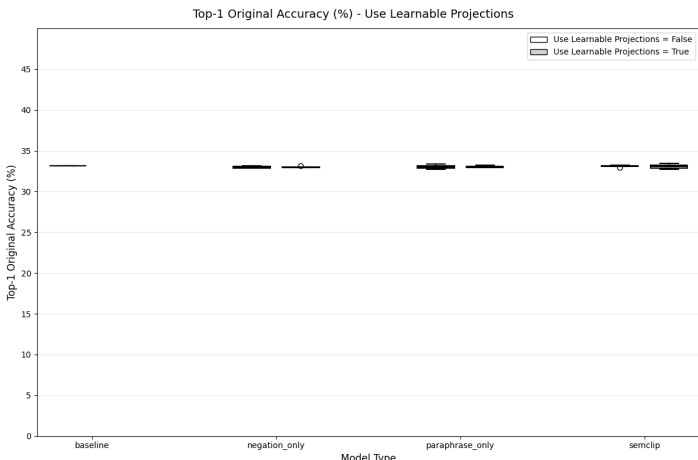

(a) Effect of setting the use of learnable projection vectors on Top-1 accuracy using original caption for image matching.

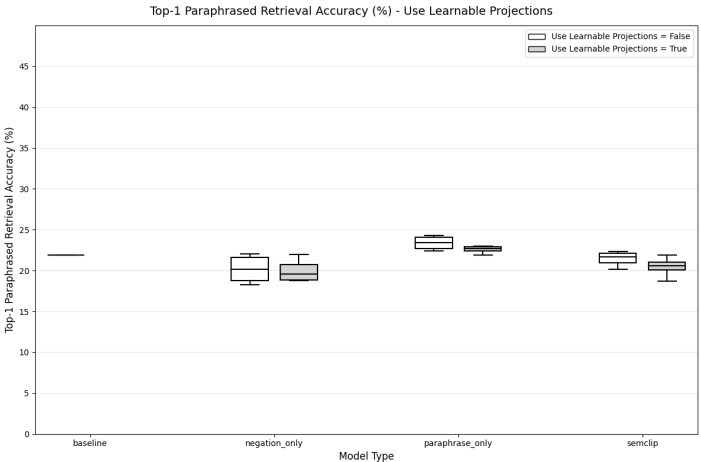

(b) Effect of setting the use of learnable projection vectors on Top-1 accuracy using paraphrased caption for image matching.

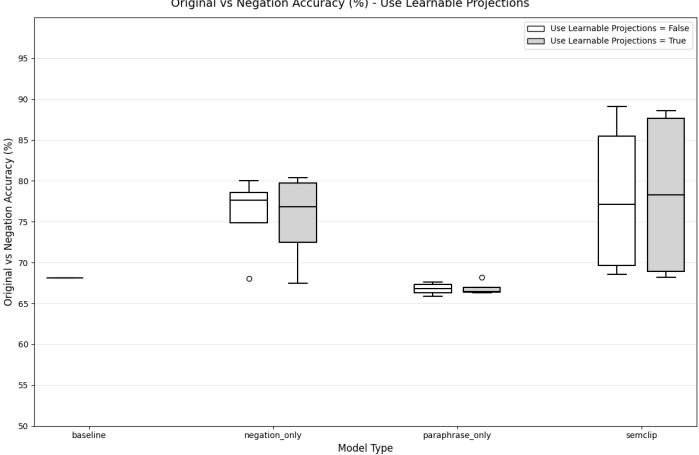

(c) Effect of setting the use of learnable projection vectors on Top-1 accuracy using original caption over negated caption for image matching.

Figure 4: Effect of setting the use of learnable projection vectors on image matching accuracies using trained model (finetuned with CC-Neg dataset (Singh et al., 2024)).

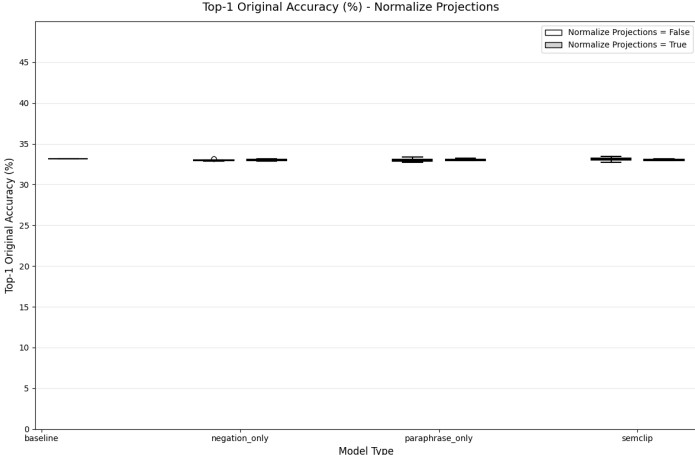

(a) Effect of setting the projection vectors normalization on Top-1 accuracy using original caption for image matching.

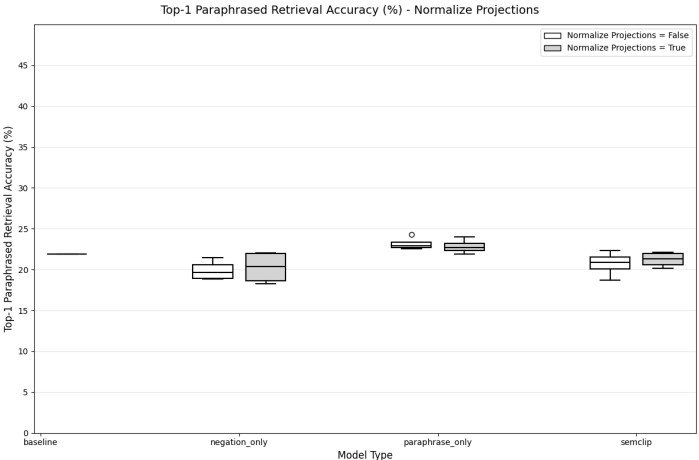

(b) Effect of setting the projection vectors normalization on Top-1 accuracy using paraphrased caption for image matching.

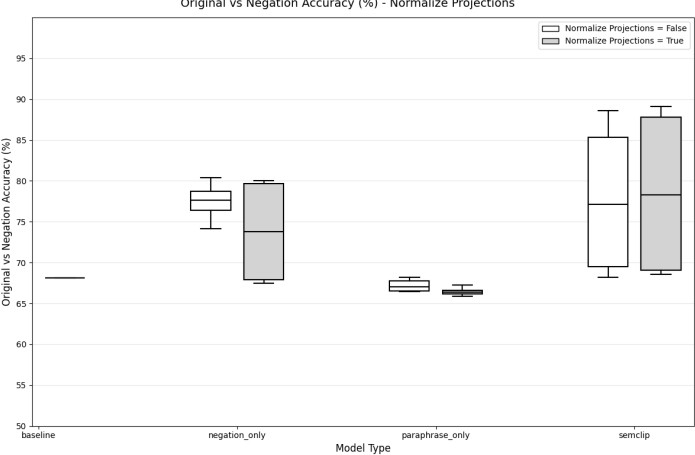

(c) Effect of setting the projection vectors normalization on Top-1 accuracy using original caption over negated caption for image matching.

Figure 5: Effect of setting the projection vectors normalization on image matching accuracies using trained model (finetuned with CC-Neg dataset (Singh et al., 2024)).

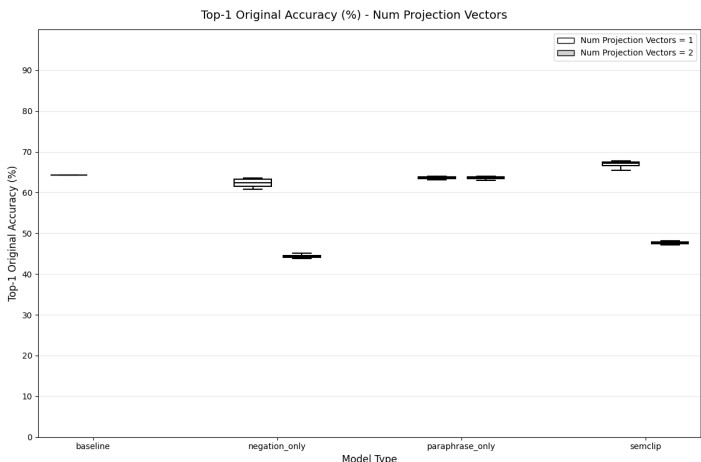

(a) Effect of setting the number of projection vectors on Top-1 accuracy using original caption for image matching.

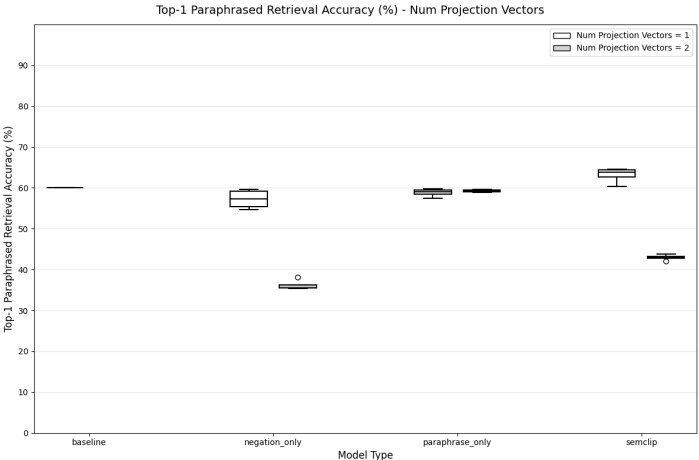

(b) Effect of setting the number of projection vectors on Top-1 accuracy using paraphrased caption for image matching.

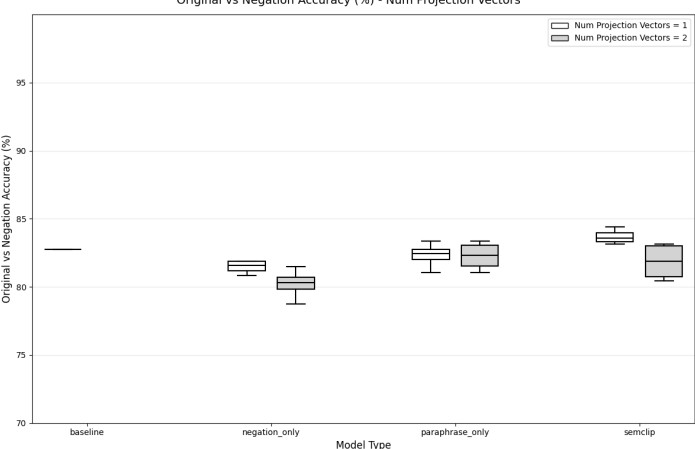

(c) Effect of setting the number of projection vectors on Top-1 accuracy using original caption over negated caption for image matching.

Figure 6: Effect of setting the number of projection vectors on image matching accuracies using trained model (finetuned with Sugarcrepe++ dataset (Dumpala et al., 2024)).

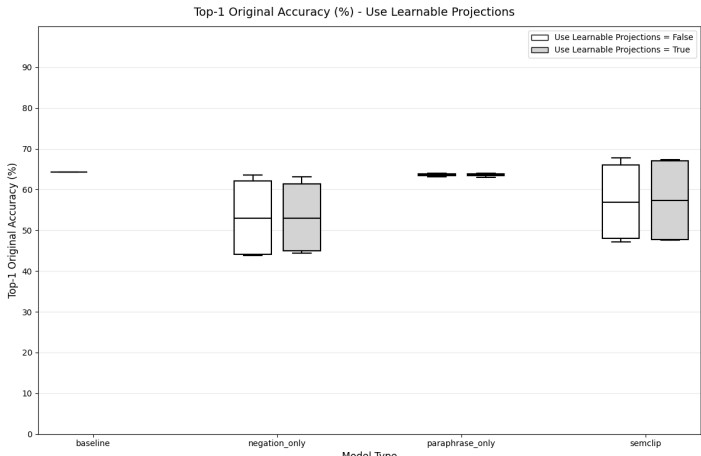

(a) Effect of setting the use of learnable projection vectors on Top-1 accuracy using original caption for image matching.

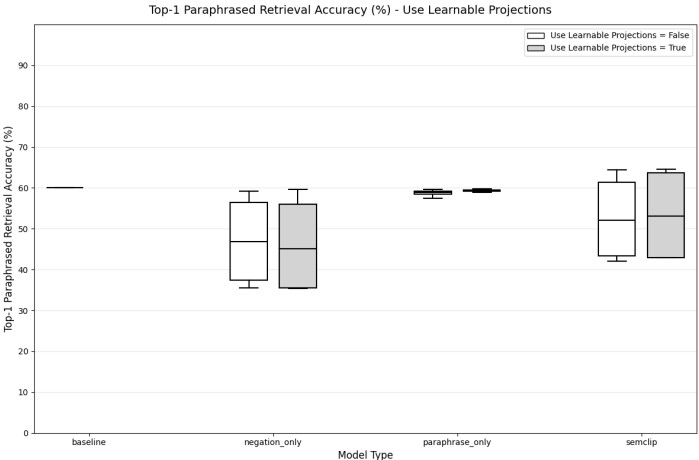

(b) Effect of setting the use of learnable projection vectors on Top-1 accuracy using paraphrased caption for image matching.

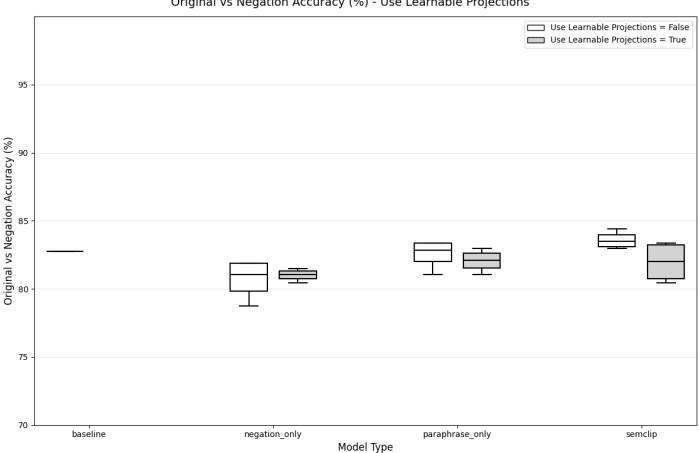

(c) Effect of setting the use of learnable projection vectors on Top-1 accuracy using original caption over negated caption for image matching.

Figure 7: Effect of setting the use of learnable projection vectors on image matching accuracies using trained model (finetuned with Sugarcrepe++ dataset (Dumpala et al., 2024)).

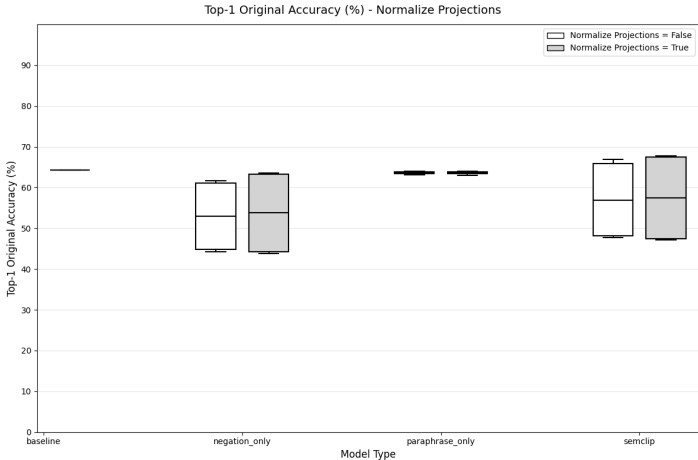

(a) Effect of setting the projection vectors normalization on Top-1 accuracy using original caption for image matching.

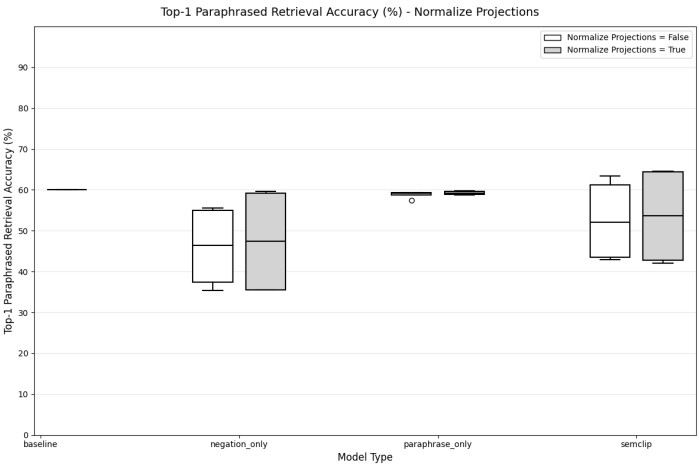

(b) Effect of setting the projection vectors normalization on Top-1 accuracy using paraphrased caption for image matching.

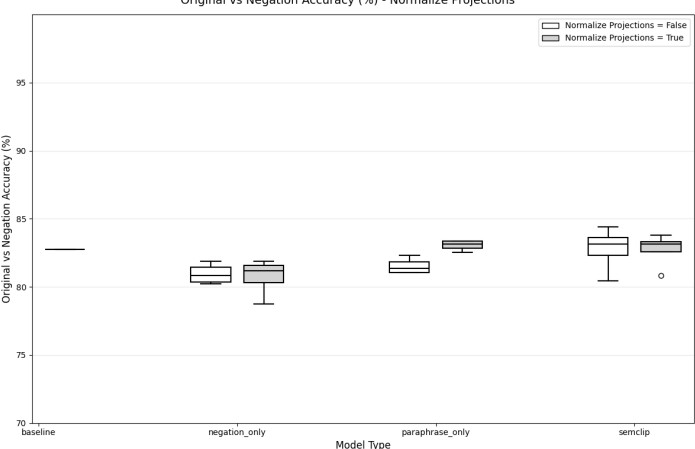

(c) Effect of setting the projection vectors normalization on Top-1 accuracy using original caption over negated caption for image matching.

Figure 8: Effect of setting the projection vectors normalization on image matching accuracies using trained model (finetuned with Sugarcrepe++ dataset (Dumpala et al., 2024)).

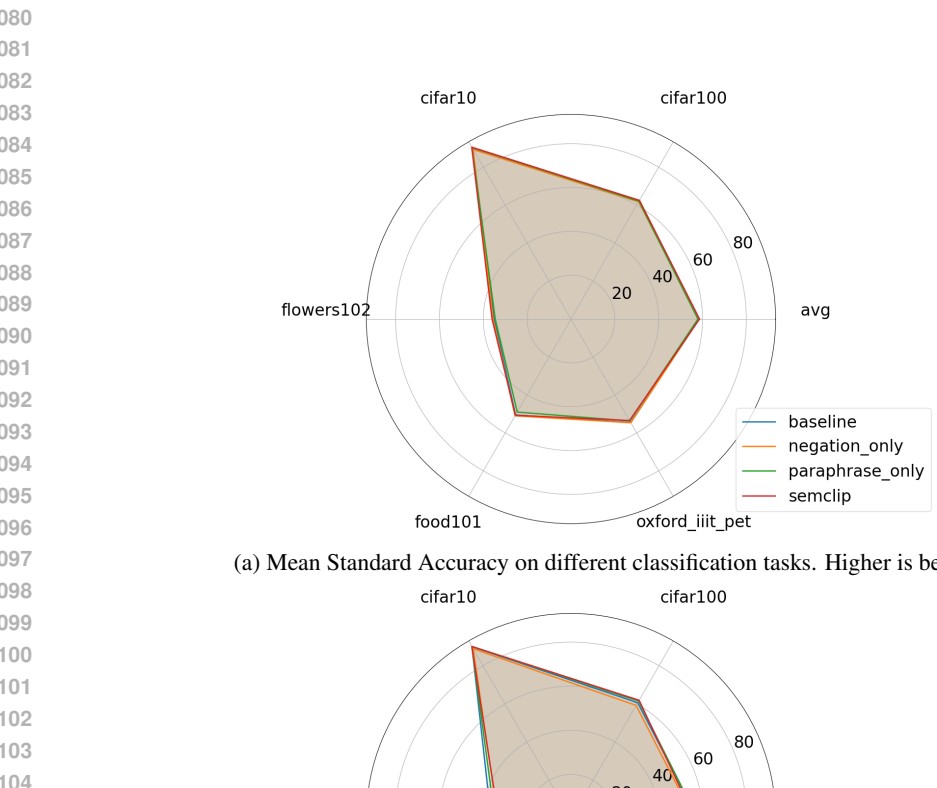

(a) Mean Standard Accuracy on different classification tasks. Higher is better.

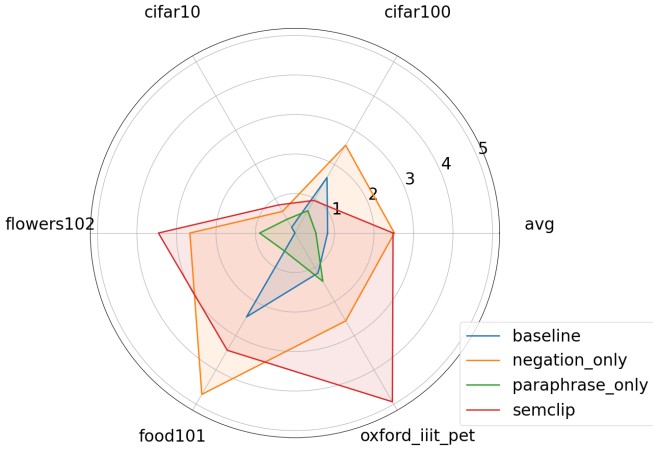

(b) Mean Negated Accuracy on different classification tasks. Lower is better.

(c) Mean Accuracy Delta on different classification tasks. Higher indicates better differentiation.

Figure 9: Mean accuracy on downstream classification tasks by models with different training loss terms (finetuned with CCNeg dataset (Singh et al., 2024)).

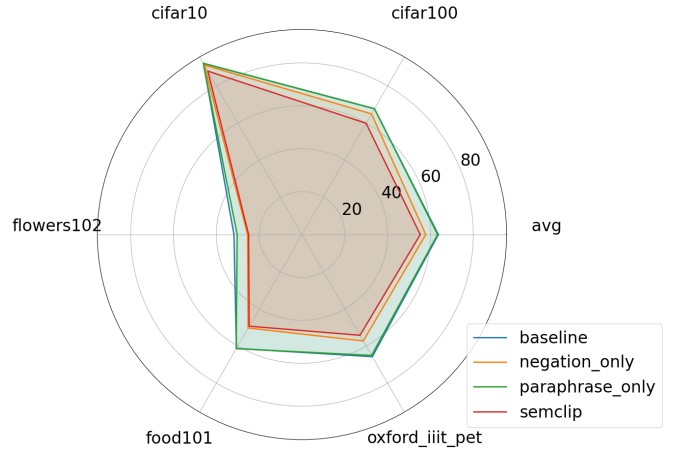

(a) Mean Standard Accuracy on different classification tasks. Higher is better.

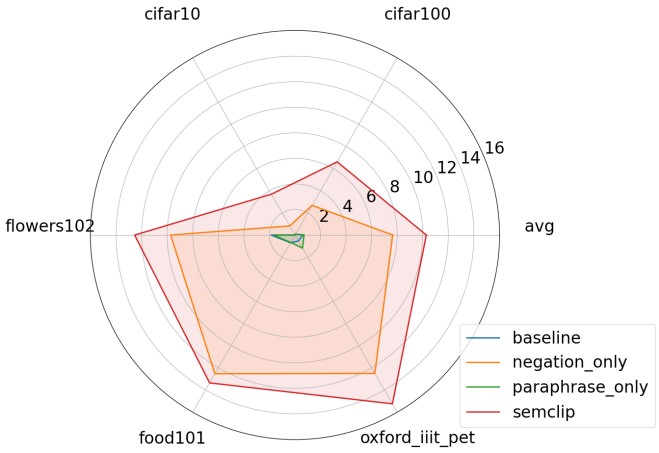

(b) Mean Negated Accuracy on different classification tasks. Lower is better.

(c) Mean Accuracy Delta on different classification tasks. Higher indicates better differentiation.

Figure 10: Mean accuracy on downstream classification tasks by models with different training loss terms (finetuned with Sugarcrepe++ dataset (Dumpala et al., 2024)).

