# OpenReview forum: "Contrastive Vision-Language Learning with Paraphrasing and Negation"
_ICLR.cc/2026/Conference — Submitted to ICLR 2026_

### Official Review · Reviewer_yYBD · 2025-10-16

**Soundness:** 2
**Presentation:** 1
**Contribution:** 1
**Rating:** 2
**Confidence:** 5

**Summary:**

**Summary:**

This paper proposes SemCLIP, an extension of CLIP that jointly models paraphrasing and negation to improve semantic robustness in vision–language learning. It introduces new paraphrase and negation losses within a low-dimensional projection subspace to align equivalent captions and separate contradictory ones. Experiments on CC-Neg and Sugarcrepe++ show SemCLIP preserves CLIP’s retrieval accuracy while improving robustness to negation and linguistic variation.

**Strengths:**

**Strengths:**

Authors tackle an important problem of negation in multimodal retrieval.

**Weaknesses:**

**Weaknesses:**

- CLIP is now outdated and many new multimodal models perform much better than CLIP. See MMEB leaderboard (V1) and the models on it.
	- Most of these models are expected to be very robust to paraphrases.
- Comparison with ConCLIP, NegCLIP and ParaCLIP missing.
- Missing Ablations:
	- What is the need for extra projection layer? Ablations need to be performed.
	- Why not use a contrastive loss with the new (anchor, paraphrase, negative). Why use two seperate losses? Ablation needs to be performed.
- Writing needs to be improved:
	- "However, large multimodal models underperformed relative to LLMs" - needs citation
	- Lines 70-72: Citation/Evaluation missing. Does clip underperform on these examples?
	- Lines 87-88: What above findings?

**Questions:**

See Weaknesses

**Details Of Ethics Concerns:**

-

---

> ### Author Response · Authors · 2025-11-21
> **Response to reviewer**
>
> Reviewer yYBD highlighted in the summary of this paper that the goal is to “jointly model paraphrasing and negation to improve semantic robustness”, that the paper “introduces new paraphrase and negation losses within a low-dimensional projection subspace to align equivalent captions and separate contradictory ones” and that “experiments on CC-Neg and Sugarcrepe++ show SemCLIP preserves CLIP’s retrieval accuracy while improving robustness to negation and linguistic variation”.
>
> Reviewer yYBD also stated that the strength of this paper is that it addresses an “IMPORTANT problem”.
>
> In our view, the weaknesses of this paper listed is merely a short list of minor corrections that can be fixed easily. We however thank the reviewer for pointing these out.
>
> Specifically, the reviewer states that: “CLIP is now outdated and many new multimodal models perform much better than CLIP. Most of these models are expected to be very robust to paraphrases”.
>
> The problem of negation, however, continues to be generally accepted as being unsolved and, in fact, the best improvements in this paper’s empirical results refer to tackling the problem of negation. In terms of relevance of CLIP, we refer to the active submissions in ICLR2026 (https://openreview.net/group?id=ICLR.cc/2026/Conference#tab-active-submissions) revealing multiple pages of submissions with the keyword “CLIP”. Many research teams continue consistently to seek innovation on top of the established original CLIP approach. Contrastive learning also continues to be highly relevant. Our framework isn’t limited to CLIP but the logic should apply to any contrastive learning model. CLIP being still one of the most prominent contrastive model to date was used for better comparison of our contribution and experimental evaluations.
>
> The reviewer states “Comparison with CoN-CLIP, NegCLIP and ParaCLIP missing. Missing Ablations: What is the need for extra projection layer? Ablations need to be performed.”
>
> Please refer to the supplementary materials for the comparison with CoN-CLIP and to our reply to reviewer 84YA. NegCLIP uses data augmentation and ParaCLIP performs LLM fine-tuning, rendering a fair direct comparison impossible as stated in the paper.
>
> We have considered the use of additional projection layers. The ablation results are reported in the supplementary materials because the differences were not statistically relevant with more projections and because ablation wasn’t that relevant to provide new insight beyond the evaluations carried out that led to the discussions provided in the main paper.
>
> The reviewer also asked “Why not use contrastive loss with the new (anchor, paraphrase, negative). Why use two separate losses? Ablation needs to be performed.”
>
> WE have considered alpha and gamma values of 0 and 1 which amount in practice to the suggsted ablation. We do indicate (also in reply to reviewer 84YA) why we have evaluated the losses separately (to analyse the extreme cases). Other values can be considered which amount in practice to using a combined loss as suggested. This is indicated as future work.
>
> Finally, the reviewer scored the presentation and contribution as poor without saying much more than that. About the presentation, the reviewer stated that “writing needs to be improved” by revising the three points (below) all of which can be resolved very easily.
>
> “However, large multimodal models underperformed relative to LLMs”
> Citation will be added in final paper.
>
> An example can be referred to Rudman et. al (2025) Forgotten Polygons: Multimodal Large Language Models are Shape-Blind.  They have stated that "Our results show that while the underlying LLMs demonstrate perfect accuracy on non-visual questions about polygon names and side counts, their multimodal counterparts fail when models have to rely on images to answer questions correctly.” This demonstrates the issue of failure in the logic connectors between the different modalities. It is thus our intention to improve on such logic connection through our proposed contrastive learning approach.
>
> "Lines 70-72: Citation/Evaluation missing. Does clip underperform on these examples?”
> Citation will be added in final paper.
>
> An example can be referred to Yuksekgonul et. al (2023) WHEN AND WHY VISION-LANGUAGE MODELS BEHAVE LIKE BAGS-OF-WORDS, AND WHAT TO DO ABOUT IT?. They have proposed an ARO (Attribution, Relation, and Order) benchmark and demonstrated that the VLM models are not capable of relational understanding, linking objects to their attributes and sensitive to order. “CLIP generally performs at chance level on positional relations”
>
> "Lines 87-88: What above findings?”
>
> We will rephrase the statement in final paper.
>
> This refers to the substantial improvements obtained by ParaCLIP with data augmentation highlighted in the previous paragrph. We will replace “above findings” with “substantial improvements obtained by ParaCLIP with data augmentation”. Thank you for pointing this out.

---

### Official Review · Reviewer_3wgX · 2025-10-27

**Soundness:** 3
**Presentation:** 3
**Contribution:** 2
**Rating:** 2
**Confidence:** 4

**Summary:**

The problem of improving CLIP training is considered. The paper introduces SemCLIP incorporating a dedicated embedding projection space and a combined loss function​, that includes components for paraphrasing (L_paraphrase​) and negation (L_negation​) alongside the standard contrastive loss (L_contrastive​). SemCLIP aims to move paraphrased captions closer to the original image embeddings while pushing negated captions further away, leading to a more robust semantic alignment between text and image. Experimental results, particularly on the CC-Neg benchmark, show that SemCLIP preserves CLIP's original performance while increasing the distance to negated captions, and this robustness extends to downstream zero-shot classification tasks.

**Strengths:**

N/A

**Weaknesses:**

1. Lack of technical novelty. It is not a new idea to finetune CLIP with negation data or paraphrasing data.
2. Lack of comprehensive evaluation. The proposed SemCLIP model was only evaluated on  two compositionality benchmarks and 5 classification benchmarks (CIFAR-10, CIFAR-100, FOODS101, FLOWERS102, OXFORD Pet). This is clearly insufficient to evaluate a CLIP model. Evaluation on more benchmarks (e.g. VTAB+ for classification, COCO/Flickr for text-image retrieval) is necessary for a solid paper.

**Questions:**

What is the cost for collecting synthetic caption data by LLMs?

---

> ### Author Response · Authors · 2025-11-21
> **Response to Reviewer 3wgX**
>
> Reviewer 3wgX questions the novelty of the paper by saying that “It is not a new idea to finetune CLIP with negation data or paraphrasing data@. We do state that in the paper! In fact, we cite prominently in the paper the existing contributions to negation and paraphrasing that have inspired our work. Our claim to novelty is also stated very clearly in the introduction of the paper: Our claim is that this paper is the first to finetune CLIP with negation data AND paraphrasing data. This claim was not refuted by any of the reviewers.
>
> We disagree with reviewer 3wgX that evaluations on “two compositionality benchmarks and 5 classification benchmarks is insufficient” and that “evaluation on more benchmarks is necessary for a solid paper”. More benchmarks may be desirable (this is probably best carried out by others) but it isn’t necessary to highlight the importance of investigating paraphrasing and negation together. All the reviewers accepted the relevance of the research. As future work, dedicated data may be collected and experimental evaluations carried out, ideally on a real application rather than more benchmarks.
>
> We thank the reviewer for pointing out further possible experimental benchmarks (since the compositionality benchmark may not be the ideal dataset: the paper makes no claim on compositionality; we just used the same datasets used by the most relevant related work).
>
> On the cost of collecting synthetic caption data using LLMs to generate paraphrased/negation captions: althought we did not track the cost specifically, the primary cost would be the compute cost of VM to call out the local LLM. Based on the current pay-as-you-go rate for H100 VM (which hosts the local LLM used), it took us about 1 week for the CCNeg with 228000 captions. So a ballpark figure of USD800 to USD1000 for that task. This can however be highly dependent on the VM spec and the region that the VM is hosted. It is in any case only a one-time cost, without impact on the overall computational cost of the proposed framework.

---

### Official Review · Reviewer_84YA · 2025-11-01

**Soundness:** 1
**Presentation:** 3
**Contribution:** 2
**Rating:** 2
**Confidence:** 4

**Summary:**

This paper, SemCLIP, proposes an extension to the CLIP framework to jointly address model robustness against two critical semantic transformations: paraphrasing (equivalence) and negation (contradiction). The approach uses a new combined contrastive loss incorporating $L_{paraphrase}$ and $L_{negation}$ terms applied to LLM-generated training triples.

**Strengths:**

The paper proposes to jointly model the two opposing yet critical semantic transformations—equivalence (paraphrasing) and contradiction (negation)—within a single unified contrastive learning framework. This approach is intriguing, establishing a necessary research direction for exploring the holistic semantic robustness of multimodal models.

**Weaknesses:**

* Despite the joint objective, paraphrase robustness does not improve: on SCPP, SemCLIP underperforms the CLIP baseline ($53.1\\%$ vs. $60.0\\%$), and on CC-Neg paraphrase it trails a "Paraphrase-only" variant ($21.0\\%$ vs. $23.0\\%$). This pattern suggests a practical tension between the attractive force of $L_{\\text{paraphrase}}$ and the repulsive force of $L_{\\text{negation}}$.

* Although negation robustness improves, it remains far from CoN-CLIP ($\text{CC-Neg Acc}_{\\text{neg}}$ $78.1\\%$ vs. $99.70\\%$), with substantial downstream zero-shot classification drops ($\approx 20$-$30$ p on Foods-101, Flowers-102, etc.). This questions the competitiveness of the projection-based loss for contradiction.

* The paper lacks a mechanistic account of how the low-dimensional projection reconciles opposing forces in the joint objective. Moreover, restricting loss weights $\\alpha, \\gamma$ to $\\{0, 1\\}$ precludes assessing trade-offs; a continuous search $0 < \\alpha, \\gamma < 1$ is necessary to demonstrate optimality and robustness of conclusions.

**Questions:**

* The paper needs a mechanistic account (e.g., visual or mathematical analysis) demonstrating how the low-dimensional projection successfully disentangles the competing $L_{paraphrase}$ and $L_{negation}$ forces, as their conflict seems to cause performance degradation on paraphrasing.

* Paraphrasing accuracy dropped below the CLIP baseline. What is the root cause of the conflict between the opposing $L_{paraphrase}$ and $L_{negation}$ forces, and how does the projection space explicitly mitigate this tension?

* Since loss weights were restricted to $\{0, 1\}$, a continuous parameter grid search is warranted to find the optimal balance.

* Given the vast gap to CoN-CLIP ($Acc_{neg}$ 78.1% vs. 99.70%), what is the fundamental limitation of the projection-based loss that prevents achieving competitive performance?

* The substantial performance drop on challenging downstream tasks (e.g., Foods 101, Flowers 102) suggests a failure in generalization. Please provide insight into why the learned semantic robustness does not effectively transfer to more complex, fine-grained visual recognition and compositional reasoning tasks.

---

> ### Author Response · Authors · 2025-11-21
> **Response to Reviewer 84YA**
>
> Reviewer 84YA acknowledges that modeling “two opposing yet critical semantic transformations” is a “necessary research direction”. This work is the first to propose a combined contrastive loss incorporating paraphrasing (noted by the reviewer as equivalence) and negation (contradiction), a novelty not refuted by any of the reviewers.
>
> The reviewer assigns a poor score despite identifying no factual errors, derivation mistakes, or methodological flaws. The critique focuses on performance trade-offs, yet our empirical evaluations explicitly discuss the tension between these opposing transformations. Our goal was not to beat the state-of-the-art via exhaustive hyper-parameter tuning on specific benchmarks, but to motivate separating negation and paraphrasing in embedding space.
>
> Regarding the request for a mechanistic account: our framework allows flexibility in dimensions and alpha/gamma values (discussed in supplementary materials). We focused on extreme hyperparameter cases (0 and 1) to highlight trade-offs and compare with closest related work. The low-dimensional projection is motivated by the simple idea (as an example of ‘left’ and ‘right’ in the Introduction and defined in Section 3.2) that the projection space should mitigate tension by separating negation cases while maintaining paraphrasing.
>
> Regarding the gap to CoN-CLIP, we would suggest against drawing fundamental conclusions based on a single metric where we cannot reproduce results due to lack of transparency. As noted, other approaches compared with CoN-CLIP also failed to match their reported accuracy.
>
> Finally, the reviewer argues that “the substantial performance drop on challenging downstream tasks (e.g. Food-101, Flowers 102) suggests a failure in generalization. Please provide insight into why the learned semantic robustness does not effectively transfer to more complex, fine-grained visual regonition and compositional reasoning tasks”.
>
> Looking at Table 2 in isolation does suggest a drop, but Figures 9 and 10 show our approach often outperforms baselines on these datasets. As stated in the paper: “It may be unrealistic to assume that the combination of negation and paraphrasing should be sufficient to address the difficult task of distinguishing the very similar-looking objects in some of these datasets. Nonetheless, these results have reinforced the significance of the proposed training loss function with paraphrasing and negation loss components as a richer way of investigating semantic robustness of image-caption pairs. Figures 9 and 10 present the classification accuracy on (a) original caption and (b) negated caption, as well as (c) the accuracy delta between the two. SemCLIP has shown an improved robustness to negation overall as it has the largest delta on three of five tasks when trained with CC-Neg dataset and on all tasks when trained on SCPP”.
>
> In contrast to 84YA’s focus on the parts of our experiments that CoN-CLIP outperforms the proposed approach (SemCLIP), Reviewer 3wgX stated “Experimental results, particularly on the CC-Neg benchmark, show that SemCLIP preserves CLIP’s original performance while increasing the distance to negated captions, and this robustness extends to downstream zero-shot classification tasks”.

---

### Author Response · Authors · 2025-11-21
**Response to Reviewers comments**

We thank the reviewers for their comments and will provide targeted and hopefully satisfactory replies to all of their questions. We would like to note that all three reviews were rather short and that the scores were in our view unjustified, for the reasons outlined next.

The reviewers acknowledge the significance of this research and do not indicate any technical issues with our methodology or errors in the derivations of our results. The low scores seem to have been given because of some nuance in our empirical results and the complaint of the lack of ablation. Our mixed results, however, have been mentioned explicitly and discussed at length in the paper. Ablation results, although not crucial, are in fact also reported in the supplementary materials to back up our main claims.

The problem of handling negation in multimodal systems is a well-known bottleneck in the practical and reliable adoption of such systems. Many recent cases of failure in reasoning became known among the general public stem from this problem of negation, which this paper addresses head on by obtaining relevant empirical result improvements even if not in all the cases we have investigated.

This paper showed the combination into a new loss function of two key concepts from existing literature, negation and paraphrasing, can produce better results towards addressing the problem of negation while it does not significantly deteriorate the results using original text. It also showed that it is not the case that results will be better all the time for the reasons given in our detailed response to the reviewers.

This paper also offers a discussion around our understanding of the reasons behind the learning of negation. By evaluating the tension that exists naturally between negation and paraphrasing, the learning task in our paper can be said to be more complex than the learning task investigated by the related work that only addresses on but not both aspects at the same time. We evaluated our approach in comparison with the most-closely related work on datasets used by that related work. This means evaluating on datasets intended for the evaluation of compositionality, even though we make no claims about compositionality in our paper. All of these were discussed in detail in the empirical evaluation section of this paper but were not picked up by the reviewers.

Performing further ablation studies would not shed new light onto the paper’s conclusions. Our results showed that the proposed combination of negation and paraphrasing makes it possible to obtain improved results. We compared our combined approach with the two existing approaches that inspired us (and some others) on more than one benchmark dataset and downstream tasks. Performing more benchmark evaluations would not give us new insight beyond what we have set up to achieve. That is not to say that further experiments would not be useful. Further experiments, especially on real applications with well-curated negation and paraphrasing data, should indeed still be an useful next step. That would be better carried out also by others that follows publication of this paper, which shoed the potential of addressing the difficult problem of negation by learning negation together with paraphrasing.

---

### Author Response · Authors · 2025-11-28
**No engagement from reviewers... we hope that the new AC will be given the time to go through the details.**

It is unfortunate that none of the reviewers has engaged with our replies. There were some important aspects to discuss to do with the quality and depth of the reviews, factual errors, and what is in our view a mismatch between the reviewers' comments and the scores given. We hope that the new AC will be given the time to go through the details rather than having to rely on the scores alone with the very large number of papers and change of AC.

---

### Meta-Review · Area_Chair_iSS7 · 2026-01-14

**Summary:**

This paper aims to improve the robustness of CLIP models on negation and paraphrasing. They do so by generating synthetic captions and proposing a new negation training objective alongside the original training objective. Results are verified on several relatively small datasets.

Three reviewers all rated this paper for rejection. However, none of them got involved in the rebuttal discussion process. After the investigation, reviewer 3wgX's review was too short and wasn’t taken into account, but AC agrees that more results on larger datasets and retrieval benchmarks like coco would make this paper stronger.

AC projected that, if the discussion process proceeded without leakage, the other two reviewers' concerns would be partially resolved, but unlikely they will reach a consensus to accept this paper. Thus, AC recommends rejection at this time.

AC understands the author's feeling of frustration when reviewers were not involved in the rebuttal. AC took a look at this paper and shared the same vision that this is an very important topic, so would encourange the authors to further improve it and submit to the next venue.

**Reviewer Concerns:**

see above

**Reviewer Scores:**

still all rejection

---

### Decision · Program_Chairs · 2026-01-26

Reject